# Theme Exploration and Sentiment Analysis of Online Reviews of Wuyishan National Park

**Wei Fu [1] and Bin Zhou [2],***

1    School of International Tourism, Anhui International Studies University, Hefei 231201, China; fwei@aisu.edu.cn
2    Donghai Academy, Ningbo University, Ningbo 315211, China
*    Correspondence: zhoubin@nbu.edu.cn

**Abstract:** The study aimed to explore the relationship and interaction between humans and nature in specific areas. Latent Dirichlet allocation topic recognition and SnowNLP sentiment analysis were used to extract the topics and analyze the sentiments from visitors' online reviews of Wuyishan National Park. The conclusions were as follows: (1) The tourists mainly expressed positive emotions toward Wuyishan National Park, and the tourists acknowledged its ecological environment and natural and cultural heritage value. (2) The tourists' comments focused on four themes: tourism activities and facilities, natural and cultural heritage value, characteristic tourism products, and tourism management and services. Natural experience was the main tourism activity in Wuyishan National Park, while cultural activities were related to the tea culture. (3) The tourist facilities, ticket and reservation mechanism, and management and services of Wuyishan National Park were the main concerns of the tourists. The study suggested that Wuyishan National Park could be transformed from a tourist destination into a comprehensive national park that provides recreational experiences and environmental education. This should be conducted by (1) developing detailed natural and cultural education and experience products and (2) improving public service functions and enhancing the public welfare of the national park.

**Keywords:** national park; online review; theme exploration; sentiment analysis; recreation

## 1. Introduction

A national park is a nature reserve that is approved by the state, and which has a rich cultural heritage and historical value [1]. In the protected area classification system that was proposed by the International Union for Conservation of Nature, national parks were classified as Class II, which permits a certain degree of human activities. They can provide "spiritual, scientific, educational, recreational, and visiting activities" that are compatible with the environment and culture [2]. In China, the Overall Plan for the Establishment of the National Park System clearly states that recreation is one of the comprehensive functions of national parks [3]. Moreover, national parks can provide tourism and leisure services for the public while simultaneously protecting the ecology, and they can provide opportunities for the public to enjoy nature and receive a natural education. Despite the great progress in China's national park system [4], there is still a lack of public understanding and awareness of national parks in China.

Most of the research on visitor behavior in national parks has focused on the traditional methods of questionnaires and interviews and has used pre-set questionnaires to collect and analyze the data [5]. These research methods can be subjective and limited, and, thus, further research is needed to validate these results. Over the past decade, technological advances have led to the widespread use of Social Networking Services, through which tourists generally share their experiences. This user-generated content (UGC) is provided at a low cost and wide range of spatiotemporal scales, and it is increasingly being used as

an alternative or supplement to traditional data sources. As previous studies have paid less attention to the needs and emotional tendencies of national park visitors that are reflected in UGC, we attempted to explore this aspect.

Wuyishan National Park is a popular tourist destination, with relatively good tourism infrastructure and supporting facilities [6]. From 2019 to 2023, the average annual number of tourists was 12.32 million in Wuyishan National Park (data from the statistical yearbook of Wuyishan City, Fujian Province). Therefore, it is representative of the recreational use of national parks in China. In light of this, this study selected Wuyishan National Park as a case study to collect tourists' comments about the park on mainstream tourism and review websites and conduct text mining and sentiment analysis. This study used UGC to analyze tourism activities and tourists' emotional tendencies in national parks and to identify the factors affecting recreation services and experience attributes in national parks [7]. The study aimed to explore the relationships and interactions between humans and nature in specific areas and put forward possible solutions for promoting the use of national parks for recreation to provide a scientific reference for the management of national parks. The findings will complement the theoretical research on national parks and could be used to enhance the complex functions and operation and management of national parks.

## 2. Literature Review and Analysis Framework

### 2.1. User-Generated Content and Network Text Analysis

The emergence of online social media has fundamentally changed the dynamic nature of online media content, and everyone can now become a content creator [7]. Thus, UGC provides a large amount of intuitive unstructured data, covering a wide range of groups and with strong authenticity, which reflects the tourist experience and satisfaction attributes of the destination [8]. Currently, some scholars have analyzed the perceptions of tourists based on UGC [9]. The studies have focused on visitor access mode [10], tourists' landscape preferences [11,12], and the spatial and temporal characteristics of tourism destination intentions [13]. In addition, UGC is widely used in the construction of tourist destination images, and some studies have discussed scenic spots [14], including the city [15], the countryside [16], and the country [17]. The influence of different scales of destination tourism images and images of major festivals and events has also been investigated [8].

Ratings, pictures, and online reviews are the main types of consumer UGC [7]. Due to the numerical nature of online scoring, it has been widely used in tourism research [18,19]. Images are the visual expression of tourists' choice of destination, and they contain rich destination information. Traditional content analysis is the most commonly used image analysis method, but it can only be used to analyze a limited number of photos. In recent years, researchers have used intelligent and automated image analysis methods, such as DeepSentiBank technology. The use of this new technology to process user-defined content (including image names, labels, and descriptions) is beneficial given the large quantity of image metadata [20].

The review text is a type of unstructured text data. The network text analysis method transforms the qualitative text material into quantitative data and quantitatively analyzes the text to make inferences about the data [21]. Some scholars suggest that these methods are more authentic and reliable than previous methods such as questionnaires and interviews. One of the most widely used text mining methods is latent Dirichlet allocation (LDA), which is used to explain hidden themes in the UGC and identify the determinants of customer satisfaction and dissatisfaction [22]. LDA has shown good performance in previous studies, and, thus, this study chose to use LDA as a topic modeling technique to analyze the reviews.

### 2.2. User-Generated Content and Network Text Analysis of the National Parks

Based on UGC, researchers have investigated the tourism image perceptions of national parks [23], including recreation perceptions [24], the construction of the National Park Brand Perception System [25], the attractiveness of recreation resources in national parks [26], and travel experiences at different scales [27]. Network text data is the main

data source that is used for the study of tourists' perceptions [28]. Chen Rongyi et al. (2020) revealed that the influencing factors of tourist satisfaction in the recreational use areas of national parks consisted of four dimensions: tourism landscapes, location and transportation, recreational experiences, and management and services [29]. Some scholars have also studied the perception of congestion in national parks [30]. Previous studies have revealed that there is an obvious difference between the experts who lead the planning of national parks and the tourists who participate in the experience in terms of the importance of the perception elements of the national parks [25].

User-generated content data (especially image data) with geo-tagged information provide continuous information about people's activities and interactions with the environment at different spatial and temporal scales, and the data have become a useful supplement to tourist monitoring tools in large areas or areas that are difficult to monitor. When compared with traditional survey data [31], UGC data is used to study the preferences of visitors in national parks [32] and behavior patterns [33]. Scholars use UGC to assess visitor sources and recreation patterns in national parks and to predict visitor sources by estimating the location of social media visitors' homes at different spatial scales [33]. Nevertheless, the data quality and representativeness can be limited due to the lack of personal information in UGC data because of personal information protection.

Based on the existing literature, text and photos are the main data sources for UGC data analyses of national parks. The research content includes the satisfaction of tourists, the preferences of national park tourists, the construction of tourism images, and the spatial and temporal distribution of tourists. Moreover, it ranges from single-problem research to multi-dimensional comprehensive research. Video data is one of the types of UGC big data, and it is widely used in tourism destination marketing practices. However, due to technical constraints, it has not yet been used in the study of national parks. Thus, although some scholars have researched national parks using UGC, further research is needed to address the limitations. The complexity of national parks requires comprehensive research methods. Qualitative research methods and quantitative research methods need to be combined. In addition, traditional investigation methods, grounded theory, importance–performance analysis, computer technology, and GIS technology should be more closely integrated. With the development and publication of new models and data sets, research methods are constantly improving, and the application of computer technology in UGC data analysis should be explored in the future.

*2.3. Analytical Framework of the Study*

In this study, the tourist reviews and travel notes regarding Wuyishan National Park on the Ctrip, Qunar, TripAdvisor, Honeycomb, and Dianping platforms were used as data sources to visualize the text data. Based on the term frequency inverse document frequency (TF-IDF) algorithm, the high-frequency words in the text were extracted, a word cloud map was drawn, and the keywords in the comment text were analyzed. The LDA topic model and sentiment analysis method were used to mine the user comments, and the actual perceived attention and satisfaction of the tourists were compared and analyzed. The analysis framework is shown in Figure 1.

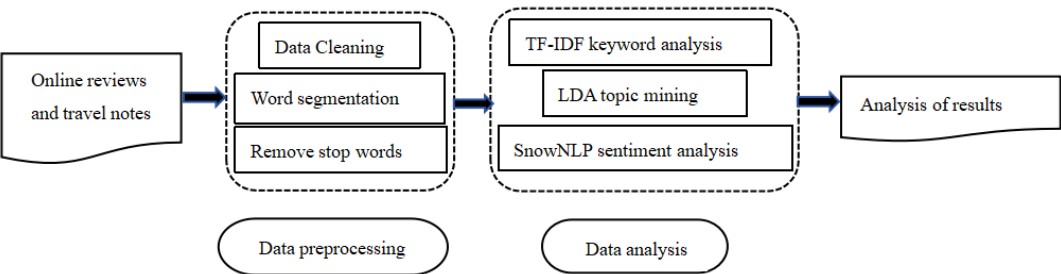

**Figure 1.** Review of the analysis framework of the online reviews of Wuyishan National Park.

## 3. Data Sources and Research Methods

### 3.1. Overview of the Focus of the Study

Wuyishan National Park spans the Fujian and Jiangxi provinces and is one of the first five national parks that were formally established in China(Figure 2). It includes the most complete, typical, and largest native evergreen broad-leaved forest ecosystem at the same latitude on Earth [1]. It has a large gene bank of species and is the origin of many types of biological specimens. The nature and culture of the Wuyi Mountains are harmonious and unified, and they have a unique landscape of "clear water and Danshan". Furthermore, they are important Buddhist and Taoist mountains in China, being the birthplace of Zhu Xi's Neo-Confucianism and oolong and black tea. Therefore, the park is a good representative of typical national parks worldwide and has a high value for natural protection, scientific research and development, and leisure and recreation [34]. Wuyishan National Park covers a total area of 1279.82 km². The Fujian Wuyishan Scenic Spot is well-known as a world cultural and natural heritage site, and it is also an important space for the zoning management of Wuyishan National Park.

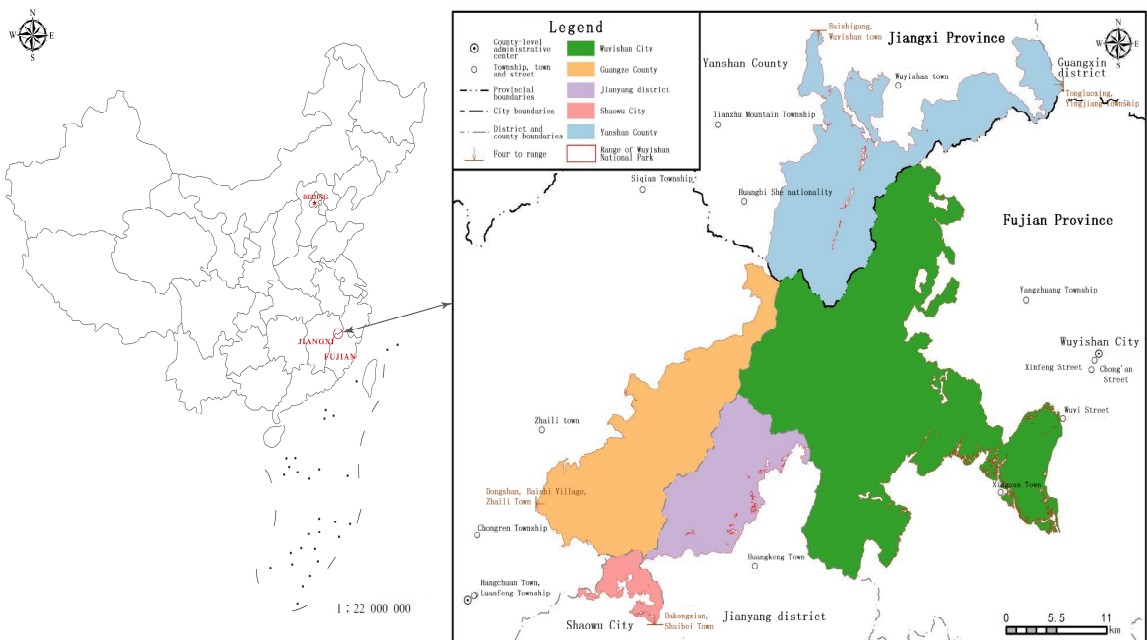

**Figure 2.** Map of Wuyishan National Park.

### 3.2. Data Acquisition and Preprocessing

To ensure that the data are comprehensive, objective, and accurate, this study selected well-known tourism portals at home and abroad as data sources: Ctrip, Honeycomb, Qunar, TripAdvisor, and Dianping (a specialized third-party consumer review website). Octopus Collector and Python 3.8.8 [35] were used to write web crawler code and collect relevant comment information about Wuyishan National Park, including user ID, comment content, comment time, score, comment praise, and other fields. To ensure the quality of the data, the collected data was preprocessed. Firstly, the repeated comments, noise data, and invalid symbols, expressions, and text were removed. After text screening, a total of 7124 tourist comments were obtained, with a total of 1,043,962 words of text for analysis. The data collection period spanned from 9 June 2008 to 18 January 2024 (Table 1). Secondly, Jieba, the most accurate and widely used Chinese word segmentation database in Python, was used to segment the data, and the word database of Harbin Institute of Technology and manual screening were used to filter the results of the word segmentation, remove stop words, blank spaces, special symbols, and meaningless words such as "you", "me", and "several", and replace synonyms such as "Dahong" and "Dahongpao" and "scenery" and "scene".

**Table 1.** Data volume statistics for the different platforms.

| Platform | Number of Valid Comments (Pieces) |
|---|---|
| Ctrip | 1680 |
| Qunar | 808 |
| Mafengwo | 49 |
| TripAdvisor | 400 |
| Dazhongdianping | 4187 |

*3.3. Research Methods*

3.3.1. Term Frequency Inverse Document Frequency Algorithm

The TF-IDF algorithm is a common weighting technique for data mining, which uses the product of term frequency (TF) and inverse document frequency (IDF) for the judgment and weight calculation of the keywords [8]. The TF represents the frequency of the keywords appearing in the document, which is usually normalized to prevent the influence of text length. The calculation formula is as follows:

$$TF_{(w)} = \frac{N_w}{N} \tag{1}$$

where $TF_{(w)}$ represents the word frequency of the word $w$; $N_w$ is the number of occurrences of the word $w$ in a document; and $N$ is the total number of entries in the document.

The IDF can be obtained by dividing the total number of documents by the number of documents containing the word and then calculating the logarithm of the quotient. The calculation formula is as follows:

$$IDF_{(w)} = \ln \frac{Y}{1 + Y_w} \tag{2}$$

where $IDF_{(w)}$ represents the inverse file frequency of the word $w$; $Y$ is the total number of documents in the text base; and $Y_w$ is the number of documents containing the word $w$ plus 1 to avoid the case where the denominator is 0 because not all documents include the word. If the number of documents containing the keyword $w$ is small and the IDF is large, it indicates that the term has a good ability to distinguish the categories [8]. The TF-IDF algorithm was used to estimate the importance of a word to the whole corpus, identify the focus of a specific text, and find the keywords of the text. The higher the TF-IDF of the word, the more important the project was to the tourists, and, thus, the project was more likely to affect the tourists' overall evaluation of Wuyishan National Park.

3.3.2. Latent Dirichlet Allocation Topic Clustering

LDA is a topic model, which can be used to identify the topics of each document in the document set in the form of a probability distribution, and each topic can also be expressed as the distribution of the words from the comments [7]. Blei (2012) argues that the advantage of LDA over other dimension extraction methods is the ability to associate documents with multiple underlying topics [36]. The LDA model can be used to identify the potential topics in large-scale document sets or corpora by clustering the hidden semantic structure of the text with an unsupervised machine learning technology, so it is widely used for the identification of user comment topics and hot spots [37]. This study constructed a topic vector containing the topic probability and the distribution probability of the corresponding words under the topic using the LDA topic clustering algorithm and obtained the topic clustering results for the Wuyishan National Park visitor reviews.

LDA was used to determine the optimal number of topics [7]. The number of topics has a direct impact on the identification of potential topics. In previous studies, domestic scholars mostly used the perplexity index to determine the quality of a prediction sample, but when the value of the perplexity index is large, it can cause model over-fitting and accurate experimental results cannot be obtained [38]. Röder et al. (2015) proposed the

coherence value of a topic, that is, to score a single topic by measuring the semantic similarity between high-scoring words in the same topic. The higher the score, the higher the topic consistency, and the better the interpretability and topic representation [39]. In this study, the indicator of topic consistency was selected to determine the optimal number of topics in the document, the LDA model in Gensim was used for topic modeling, and the preprocessed tourist comments were clustered to mine the related topics and topic features. Before implementing the model, the theme number (K) was determined by calculating the theme consistency value.

### 3.3.3. Text Sentiment Analysis

Text sentiment analysis, also known as opinion mining, refers to the mining and analysis of the subjectivity, objectivity, opinion, emotion, and polarity of the text through computing technology, and it can be used to classify the emotional tendency of the text, which is important in the field of natural language understanding [40]. This method is widely used to mine the emotional tendencies that are expressed by words in online news, reviews, blogs, microblogs, and social networks, and its development is consistent with the development of social media [40]. The method of sentiment analysis that was adopted in this study was SnowNLP, which uses the naive Bayesian principle to train and predict the data, and is easy to operate and implement [41]. It is a Chinese natural language processing Python library, which can be used for the sentiment analysis of Chinese corpora. The scores range from 0 to 1. Emotional tendencies are divided into positive and negative emotions. The closer the score is to 1, the more positive the attitude that is expressed by the text, and the closer the score is to 0, the more negative the attitude that is expressed by the text.

## 4. Research Results

### 4.1. Keyword Analysis Based on the Term Frequency Inverse Document Frequency Algorithm

To intuitively show the focus and theme of the user comments, the top 50 words with the highest TF-IDF value after the comment word segmentation were extracted (Table 2) and the word cloud library in Python was used to draw the word cloud map (Figure 3).

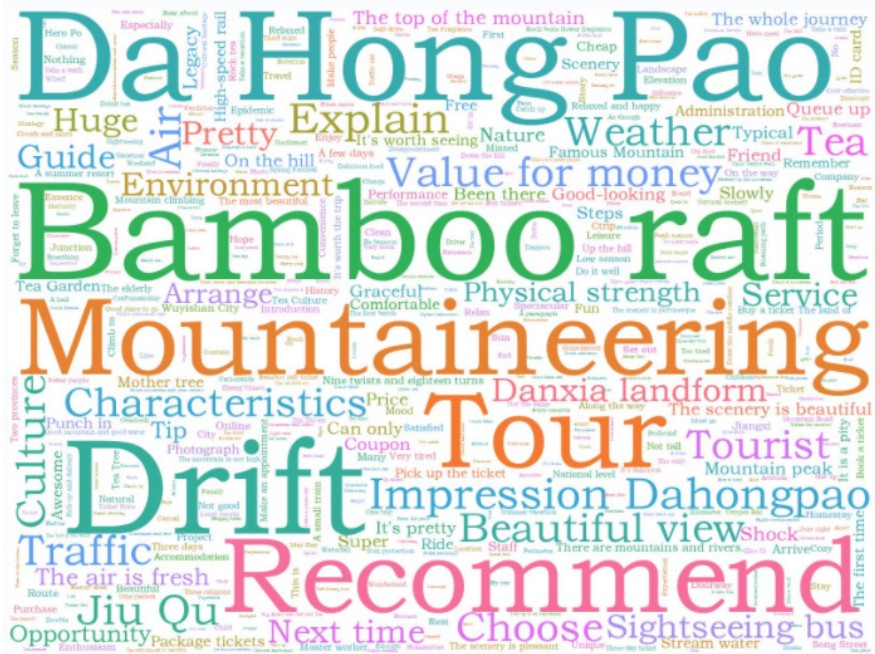

**Figure 3.** Word cloud map based on the term frequency inverse document frequency (TF-IDF) values (the size of the text in the map corresponds to the TF-IDF value).

**Table 2.** Top 50 keywords and term frequency inverse document frequency values.

| Serial Number | Keyword | TF-IDF Value | Serial Number | Keyword | TF-IDF Value |
|---|---|---|---|---|---|
| 1 | Scenery | 0.03056 | 26 | Tourist | 0.00697 |
| 2 | Bamboo raft | 0.02615 | 27 | Choose | 0.00684 |
| 3 | Drift | 0.02270 | 28 | Sightseeing bus | 0.00678 |
| 4 | Da Hong Pao | 0.01957 | 29 | Danxia landform | 0.00668 |
| 5 | Mountaineering | 0.01429 | 30 | Beautiful | 0.00668 |
| 6 | Time | 0.01409 | 31 | Service | 0.00662 |
| 7 | Recommend | 0.01296 | 32 | Hotel | 0.00641 |
| 8 | Ticket | 0.01295 | 33 | Happy | 0.00635 |
| 9 | Tour | 0.01269 | 34 | Next time | 0.00634 |
| 10 | Experience | 0.01225 | 35 | Environment | 0.00619 |
| 11 | Impression Dahongpao | 0.01002 | 36 | Huge | 0.00618 |
| 12 | Air | 0.00983 | 37 | Travel | 0.00615 |
| 13 | Traffic | 0.00926 | 38 | Beautiful scenery | 0.00609 |
| 14 | Fun | 0.00918 | 39 | Pretty | 0.00603 |
| 15 | Explain | 0.00889 | 40 | Guide | 0.00589 |
| 16 | Value for money | 0.00827 | 41 | Interesting | 0.00586 |
| 17 | Jiuqu | 0.00826 | 42 | Physical strength | 0.00576 |
| 18 | Nature | 0.00788 | 43 | The scenery is beautiful | 0.00574 |
| 19 | Culture | 0.00769 | 44 | Arrange | 0.00572 |
| 20 | Landscape | 0.00757 | 45 | Comfortable | 0.00559 |
| 21 | Weather | 0.00755 | 46 | Opportunity | 0.00554 |
| 22 | Characteristics | 0.00743 | 47 | The air is fresh | 0.00552 |
| 23 | Tea | 0.00711 | 48 | Tip | 0.00551 |
| 24 | Performance | 0.00703 | 49 | The top of the mountain | 0.00542 |
| 25 | Beautiful view | 0.00701 | 50 | Shock | 0.00541 |

Firstly, based on the distribution of the keyword TF-IDF values, the tourists were most concerned about the natural landscape and experience activities of Wuyishan National Park. The TF-IDF values of "Scenery", "Bamboo raft", "Drifting", "Dahongpao", "Mountaineering", and "Experience" were all in the top ten, and the related keywords "Nature", "Culture", and "Landscape" were also at the forefront. Additionally, "Dahongpao" had a dual reference, which referred to the impression of the Dahongpao live performance and the taste experience of the famous tea "Dahongpao". The tourists were interested in original ecological and experiential tourism products, and Wuyishan National Park met the needs of the tourists for experiences and sightseeing. Secondly, "Ticket", "Transportation", "Explanation", "Sightseeing bus", "Service", "Hotel", "Guide", and other high-weight words represented the reception facilities and services, such as the transportation facilities (Sightseeing bus), accommodation facilities (Hotel), personnel services (Explanation, Guide, and Tip), which showed that the reception facilities and services were important for tourists. Thirdly, "Characteristics", "Beautiful", "Beautiful scenery", "Pretty", "The scenery is beautiful", and "The air is fresh" reflected the recognition of the natural landscape and natural environment of the park. The protection of the ecological environment and resources is the basis for the main management goal of implementing the concept of public welfare and providing recreational services in Wuyishan National Park. Fourthly, words such as "Recommendation", "Fun", "Happy", "Interesting", and "Comfortable" reflect that the emotional tendency of the tourists had a higher weight, reflecting the higher recognition and satisfaction of the tourists in Wuyishan National Park.

*4.2. Hot Topic Extraction of the Comments*

According to the consistency calculation, the number of topics for the relevant text achieved the highest consistency when K = 4 (Figure 4). Therefore, this study chose K = 4 as the target number of topics for unsupervised learning to train the model, and it was iterated 300 times.

The results of the topic modeling were visualized by the PyLDAvis library (Figure 5). "Dahongpao", "Drifting", and "Mountaineering" were the words with the highest frequency of interaction among the themes because they represent the main tourism content in Wuyishan National Park.

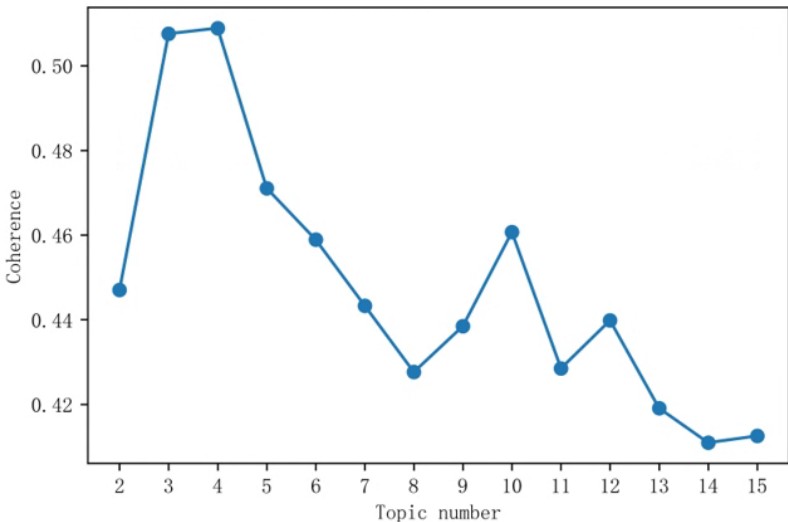

**Figure 4.** Coherence score.

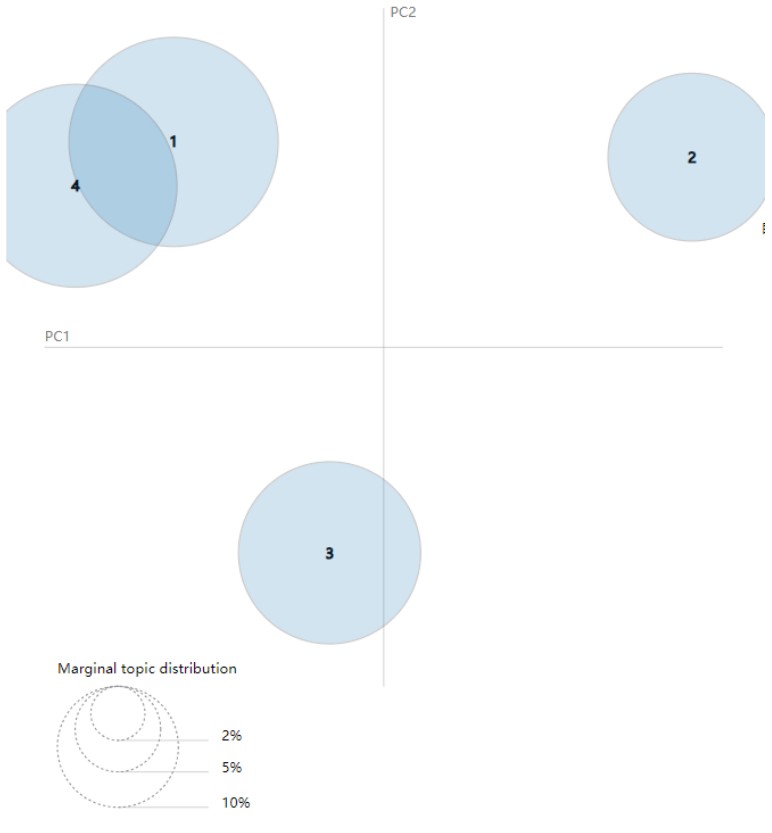

**Figure 5.** Visualization of the PyLDAvis results (the circles represent the topics, the distance between the centers of the circles indicates the semantic distance of the topics, the area of the circles indicates the extent to which they are discussed in the comments, and the greater the overlap between the circles, the more mutual terms between the topics [22]).

According to the results of the LDA clustering, the feature words of each theme were extracted according to the order of probability of occurrence, and the specific meaning of each theme was summarized to obtain the four themes, namely tourism activities and facilities, natural and cultural heritage value, characteristic tourism products, and

tourism management and services (Table 3). In terms of (1) tourism activities and facilities, there were high-frequency words such as "Dahongpao", "Time", "Drifting", "Hotel", "Mountaineering", "Travel", "Sightseeing bus", "the scenery of Wuyi Mountain is really beautiful but it's really tiring! You have to take a scenic bus", and "there are several ways to get to the Impression Dahongpao Theater, taxi, tricycle, and bus". The main types of tourism activities in Wuyishan National Park were natural and cultural experiences, such as mountain climbing activities, watching the impression of Dahongpao live performance, and the Dahongpao tea cultural experience. Due to the scattered scenic spots in Wuyishan National Park, tourists are highly dependent on the infrastructure and service facilities, otherwise, the physical strength, energy, and time that are required to complete the tourism activities are too high. The travel planning and reception facilities of the tourism activities had an important impact on the overall experience quality of the tourists.

**Table 3.** Topic classification and feature words.

| Topic Number | Topic Name | Feature Words |
|---|---|---|
| Topic1 | Tourism activities and facilities | Dahongpao, Time, Drift, Ticket, Hotel, Mountaineering, Experience, Trip, Choose, Arrange, Recommend, Tour, Impression Dahongpao, Price, Service, Online, Sightseeing bus, Traffic, Tour guide, Explain, Performance, Cheap, Weather, Bus, Free travel, Characteristics, Happy, Physical strength |
| Topic2 | Natural and cultural heritage value | Culture, Nature, Danxia landform, Typical, Legacy, Famous mountain, Dahongpao, Drift, Natural reserve, A summer resort, History, Zhu Xi, Protect, Temple, Confucianism, World heritage, Perch, Beautiful, Unique, Ecosystem |
| Topic3 | Characteristic tourism products | Ctrip, Dahongpao, Drift, Tour, Air, Awesome, Landscape, Mountaineering, Opportunity, Tea garden, Jiuqu, Interesting, Pretty, Beautiful, Time, Fun, Tea, On the hill, Enjoy, Value for money, Beautiful view, Recommend, Impression Dahongpao, Appreciate, Nature, Tea Tree, Fresh, Scenery, Stream |
| Topic4 | Tourism management and services | Drift, Time, Mountaineering, Dahongpao, Explain, Guide, Jiuqu, Ticket, Sightseeing bus, Tourist, Tip, Fun, Recommend, ID card, Staff, The top of the mountain, Weather, Pick up the ticket, Administration, Air, Steps, Up the hill, Service, Tour, Coupon, Experience, Choose, Price, Queue up, Good-looking |

In terms of (2) natural and cultural heritage value, this theme focused on high-frequency words such as "Culture", "Nature", "History", "Protect", "Unique", and "Ecosystem". It highlighted the natural and cultural values of the Wuyi Mountains, including the Danxia landform, natural heritage, and Dahongpao tea culture, which were also mentioned in the commentary text. The comments included "boat coffin on the cliff", "Ziyang Academy founded by Zhu Xi", "Imperial Tea Garden of [the] Yuan Dynasty", and "Qixiujia Southeast", which shows that the tourists have a high degree of recognition of the ecological environment and natural and cultural heritage value of Wuyishan National Park. Therefore, its natural and cultural resources should be protected, and its ecological advantages should be maintained. In terms of (3) characteristic tourism products, the theme focused on the unique natural experience activities of the Wuyi Mountains, which were represented by "Drifting", "Mountaineering", "Landscape", and "Tea garden". Tourist books and the tourism products of Wuyi Mountain National Park were recommended through online travel agent (OTA) platforms, such as Ctrip. Currently, the tourism products of Wuyishan National Park are mainly sightseeing, and the participatory experience activities are mainly bamboo rafting in Jiuqu River and the tea culture exhibition in Wuyishan. However, there is still a lack of communication on the value of the natural and cultural resources in Wuyishan National Park based on the existing tourism products, and the types of natural education,

cultural experiences, and communication products for the public are limited, so these activities could be further developed.

In terms of (4) tourism management and services, the main activities were "Drifting" and "Mountaineering", which were associated with "Service" and "Management". There were also many related words, such as "Explanation", "Guide", "Ticket", "Staff", and "Queue up". From the original user reviews, it can be seen that the level of management and services affects the quality of tourists' travel experiences. For example, the comment "[the] Jiuquxi explanation is really far-fetched, without any substance".

Overall, natural experiences are the main tourism activity in Wuyishan National Park. The tourists have a high degree of recognition of the natural and cultural heritage value of Wuyishan National Park but the dissemination of the concept of "ecological protection first, national representativeness, and public welfare of the whole people" has not been clearly communicated to the tourists. This is because the existing tourism products of Wuyishan National Park are mainly sightseeing. For a national park with a large geographical area and scattered scenic spots, the level of tourism facilities, management, and services has become the focus of attention of the tourists.

### 4.3. Comment Text Topic Sentiment Analysis

Sentiment analysis of the online comments of visitors to Wuyishan National Park can help with understanding the emotional tendencies of the visitors. The SnowNLP emotion analyzer in Python was used to calculate the emotional value of the comments, and the distribution of the tourists' emotional tendencies in Wuyishan National Park is shown in Figure 6. The overall average emotional value of the comments was 0.7204, indicating that the tourists had a positive emotional attitude toward Wuyishan National Park, and most of the tourists were satisfied with it. Thus, Wuyishan National Park met their expectations.

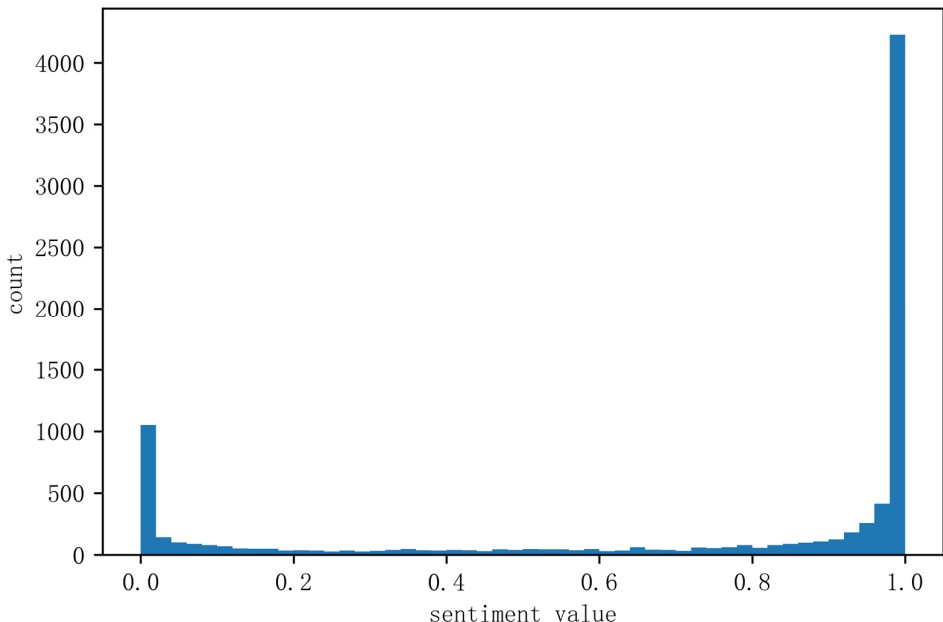

**Figure 6.** Histogram of the emotion score distribution.

It can be seen from Figure 6 that there were more extremely positive comments with emotional scores greater than 0.8 and more extremely negative comments with scores less than 0.2, with relatively few comments between the two extreme emotions. This indicates that tourists have a clear emotional tendency toward Wuyishan National Park. The tourists' positive emotional tendency toward Wuyishan National Park mainly originated from (1), the excellent ecological environment and unique natural landscape resources. Wuyishan National Park has a national scenic spot, which is a typical Danxia landform landscape featuring "Clear water and Danshan Mountain". Due to "the strictest protection" in China's

national parks, the environmental indicators of the air, water, and soil in Wuyishan National Park have reached the first-class standard of a first-class functional area, the ecological environment quality in the region has been improved, and the authenticity and integrity of the ecosystem have been effectively protected [42]. The comments mentioned that "Wuyi Mountain is extremely high in negative oxygen ions" and "the scenery here is super eye-pleasing! Everywhere you look is green." Thus, Wuyishan National Park has the advantage of natural ecological resources, and its ecological environment and the unique landscape resources should continue to be protected.

Regarding (2) the characteristics of cultural heritage, as a world natural and cultural heritage site, Wuyishan National Park has a strong cultural heritage and tea culture, and the Neo-Confucianism and religious cultures are the most prominent manifestations of its characteristic culture. The commentary mentioned "[that] the history of Wuyi rock tea can be traced back to the Shang and Zhou Dynasties, with rock rhyme (rock bone flower fragrance)" and "holy land of three religions". Therefore, Wuyishan National Park enriches tourists' cultural experience and promotes the spread of culture by creating characteristic cultural exhibitions and experience activities.

Additionally, (3) the comments for the convenient tourist traffic mentioned that "the airport can be reached by taxi in half an hour", "the sightseeing bus in the scenic area connects all the scenic spots in series", and "the high-speed rail to Wuyishan East Station [and] the bus to the scenic area [are] still very convenient". These comments indicate that the traffic inside and outside of Wuyishan National Park is convenient, and the tourists' emotional evaluation of the tourist traffic was mainly positive. Good transportation can attract more tourists, thus increasing the popularity and attractiveness of Wuyishan National Park.

The negative comments pointed out the shortcomings of the recreational utilization of Wuyishan National Park, which should be noted. The negative emotional tendency of the tourists mainly came from the following aspects. In terms of (1) consumption and service management, the tourists were dominated by negative emotions regarding shopping consumption, such as "[the] jewelry in [the] souvenir shops [are] fake goods", "no characteristics", and "not meeting [my] needs". The evaluation of "price", "deception", and "slaughter" in terms of catering consumption reflected its inadequacy in Wuyishan National Park. In terms of tourism service management, there were negative comments such as "bad attitude of [the] staff", "wild tour guide", and "too crowded", which showed that there is still much room for improvement in the recreation services and management of Wuyishan National Park.

Regarding (2) the ticket and reservation mechanism, the Wuyishan Scenic Area is the main open space in the current national park zoning. Thus, visitors need to pay to enter. The admission price is 140 yuan per person for a full ticket and 70 yuan per person for a half ticket in the peak season (March to November); 120 yuan per person for a full ticket and 60 yuan per person for a half ticket (3-day ticket) in the off-season (December to February) [5]. The sightseeing buses and activities in the area (Jiuquxi bamboo raft rafting and Impression Dahongpao performance) are booked and charged separately, with a total price of more than 300 yuan per person. The comments included "ticket prices are too high", "bamboo raft reservation is difficult", and "[it is] unreasonable for small babies to buy [bamboo raft] tickets", which confirm the tourists' dissatisfaction with the ticket pricing and reservation mechanism. In addition, the main channels for tourists to book tickets are official WeChat public numbers, OTA platforms, and travel agencies, but "changeable weather" and "too far [of a] distance [and] too long [of a] travel time leading to changes in [our] plans" were found to affect the travel arrangements of the tourists. Moreover, the communication between the booking platforms and the scenic spots was not smooth when collecting and refunding tickets. This caused obvious negative emotions, such as "package pit" and "handling fee for refund". This shows that there is still a gap between the needs of the public and the ticket fees and reservation mechanism of Wuyishan National Park, and public welfare is not fully reflected. (3) In terms of the supporting facilities, although the

tourists' emotional evaluation of tourism traffic was mainly positive, due to the mountain roads and the long distance between the scenic spots and related supporting facilities, several issues were noted, such as "few selling points in scenic spots", "unreasonable step construction", "unclear signs in scenic spots", and other issues. Furthermore, the negative emotions included "insecurity", tourists being physically "tired", and "regret".

## 5. Discussion

The findings of this study are of great significance to the theoretical research and practical development of national park tourism. Firstly, in contrast to previous research on tourist experiences, which relied more on questionnaires and interviews, this study used UGC. The data of the Wuyishan National Park visitors' online comments were objective, substantial, and chronological, covering the Wuyishan Scenic Spot, Wuyishan National Park System Pilot, and Wuyishan National Park, which provides a theoretical foundation for a follow-up study on tourism image construction and the realization of recreational functions in the construction process of Wuyishan National Park.

Secondly, the thematic clustering analysis of the comments and sentiment analysis showed that the following:

(1) That the tourists' emotions toward Wuyishan National Park were positive overall, and they had a high recognition of the ecological environment and natural and cultural heritage value of Wuyishan National Park. This conclusion is consistent with the findings of Tong Bisha and Chen Guangpu (2020) on the tourism image perceptions of pilot units of national parks in China [23]. This is closely related to the nature of national parks. National parks are important natural ecosystems, and they protect the most unique natural landscapes and the most abundant biodiversity in China [43]. They represent the core characteristics of national nature and culture and have a high value for natural protection, scientific research and development, and leisure and recreation [33]. The main aim of the construction of national parks is to realize the scientific protection and rational utilization of resources. Therefore, standardizing the management of resources and the environment in national parks and carrying out recreation and educational utilization that are related to protection are the key points for the future development of recreation in national parks.

(2) The natural experience was represented by mountain climbing and bamboo raft rafting in Jiuqu River, which were the main tourism activities of Wuyishan National Park. The cultural experience activities mainly included the "Impression Dahongpao" live performance, tea-picking, tea-making, and tea-tasting of Wuyi rock tea. The tourists mentioned these activities the most, but due to the constraints of the organization and management, the tourists had different opinions on the quality of the experiences. Notably, although the Wuyi Mountains are rich in cultural heritage, the internal cultural landscape spaces are far away from each other, and the continuity is poor [44]. There were few local historical and religious cultural experience activities, which was not conducive to the protection and dissemination of cultural heritage. This is basically consistent with the research conclusions of Chen Rongyi et al. (2020) [29]. The unique ecological environment or rich cultural landscape of national parks are great attractions for tourists, but tourists often do not get to experience nature fully due to the limitations of human resources, tourism activities, and route design and management. Additionally, to keep the ecosystem intact, the professionalism of managers limits the tourists' appreciation, exploration, and understanding of nature and local culture [45]. Therefore, strengthening research on the scientific and cultural values of national parks and highlighting the cultural, scientific, ecological, and environmental education experiences that can be conducted in the process of recreation should be an important focus in the future.

(3) The tourist facilities, ticket and reservation mechanism, and management and services of Wuyishan National Park have become the focus of tourists, and their short-comings affect the satisfaction of the tourists. These findings were supported by Dou Yaquan [5], and the emotional tendency of the tourists toward the scenic facilities, services, and management in this study was consistent with the findings of Yan Yaoyao (2020) in

the Great Wall National Park [46] and Huang and Wang (2021) [24] in Pudacuo National Park. This shows that these are common problems in China's national parks. Some of the fundamental purposes of the establishment of public welfare national parks are setting them up for public interest, charging low fees to the public, educating the public, and allowing the public to actively participate [47]. However, China has not yet formed an effective mechanism for the operation and management of national parks [48]. This has led to many restrictions on ecosystem integrity maintenance, visitor behavior, park zoning, franchise projects, cooperative co-management, and other aspects, which affects the willingness of the citizens to participate in national park tourism activities and experience natural and cultural education and hinders the realization of public welfare in national parks. Thus, there is a need to determine how to fully implement the concept of China's national park system.

There were some limitations to this study. Firstly, the sources for the tourist reviews were limited, and the collected samples were restricted to online reviews of tourist destinations. Secondly, there is a certain degree of subjectivity in the induction of thematic clustering in the process of research, and how to eliminate the influence of subjective will should be the focus of future in-depth research. Thirdly, China's national parks have been established for a short time, and their management system, concept, planning, and construction are still in the exploratory stage. Therefore, tourists are not clear about the functions of national parks, which leads to low attention to the concept of national parks in tourists' comments.

## 6. Conclusions and Future Recommendations

### 6.1. Conclusions

This paper uses Jieba word segmentation technology to preprocess and extract features from the online comments of tourists on major tourism OTA platforms and social networking sites at home and abroad about Wuyishan National Park. Then, the LDA method is used to cluster these comments to reveal the different themes that tourists pay attention to in Wuyishan National Park. At the same time, through the training of a large body of text, in order to more accurately identify the hidden emotional tendencies in the comments, the SnowNLP library in Python is used for emotional analysis to distinguish the emotional emotions expressed in the comments, such as positive, negative or neutral. The study found the following: (1) Tourists mainly focus on the tourism activities and facilities, natural and cultural heritage value, characteristic tourism products, and tourism management and services of Wuyishan National Park. Among them, natural experience is the main tourism activity of Wuyishan National Park. Cultural experience activities revolve around tea culture, with less local history and religious culture experience activities. (2) Tourists' overall emotion for Wuyishan National Park is positive. Tourists believe that Wuyishan National Park has an excellent ecological environment, unique natural landscape resources, significant cultural heritage characteristics, and convenient internal and external tourism transportation. However, some negative emotions mainly come from consumption and service management, supporting facilities, and the tickets and reservation mechanism. Affected by this, the public welfare concept of the external national park is not fully reflected, which is the direction that needs to be strengthened and improved in the future.

### 6.2. Future Recommendations

Based on the above conclusions and discussions, this paper has suggested specific policy recommendations for the improvement of the future tourism development and management of Wuyishan National Park.

1. Further development of natural and cultural education and experience products

Firstly, natural and cultural landscape resources should be utilized, such as ancient pedaling roads, ancient tea gardens, and ancient relics. The historical celebrities, tea allusions, folk activities, and traditional tea-making techniques should be combined with landscape relics to create highly recognizable tourism activities with regional cultural

characteristics [28]. To enhance the awareness of religion, history, and agricultural culture among tourists, the existing Wuyi rock tea experience activities could be expanded, animal and plant research experience products could be designed [49], and immersive historical and cultural exhibitions could be developed. Moreover, according to the spatial and temporal distribution pattern of the landscape, characteristic natural and cultural sightseeing routes and related thematic activities could be created by connecting the features and activities [44]. Secondly, environmental education could be improved through printed materials (such as picture albums and popular science books in parks) and museums and environmental explanations (such as popular science lectures in national parks and campus activities and natural classes) to transform the park from a tourist destination to a national park that provides a comprehensive recreational experience and environmental education that highlights the long-term interactive value of natural ecology and culture [5].

2.  Improving the public service function and enhancing the public welfare of national parks

National parks are special public goods that are owned by everyone, so low or appropriate admission prices are required [23]. Wuyishan National Park is one of the five national parks in China with relatively high entrance fees and activity charges. It is recommended that the population and dates that are covered by free or preferential entrance policies should be expanded, and annual tickets should be introduced for national parks. In addition, the design of existing joint tickets and the booking mechanism should be improved, the sales on different booking platforms should be standardized, and ticket booking should be well-linked with the franchise activities so that tourists can truly experience the convenience of price diversification. Additionally, the services and management of the park should be improved, and the traffic management system should be used to improve traffic convenience for tourists. Furthermore, the intelligent guide system of the park should be continuously improved through mobile applications, augmented reality guides, and live broadcasting. This could be used to provide more accurate and detailed information for tourists, market natural phenomena or cultural connotations, help tourists positively experience Wuyishan National Park, enhance national pride and identity, and improve public interest and awareness of environmental protection. Lastly, the services and facilities for the park visitors should be improved, such as the toilets, vending machines, and rest stations, and the training and management of the staff should be strengthened. The recommendations of this study could be used to improve the development of tourism products and enhance the public service function and public welfare of national parks [23].

**Author Contributions:** Conceptualization, methodology, data curation, and writing—original draft preparation, W.F.; supervision, writing—review and editing, and formal analysis, B.Z. All authors have read and agreed to the published version of the manuscript.

**Funding:** This study was supported by the National Natural Science Foundation of China (Grant No. 42171223).

**Data Availability Statement:** Data are contained within the article.

**Acknowledgments:** The authors gratefully acknowledge the support of the funding. We also sincerely thank reviewers and the journal editors.

**Conflicts of Interest:** The authors declare no conflicts of interest.

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
