# Peer review of "Theme Exploration and Sentiment Analysis of Online Reviews of Wuyishan National Park"

_land, doi:10.3390/land13050629_

Round 1
Reviewer 1 Report
Comments and Suggestions for Authors
Good presentation of basic constructs of the research within the abstract, together with presentation of the main findings, followed by adequate choice of the key words.
Generally, this is an interesting paper with focus of the research aimed at providing the results for better understanding the relationship and interaction between humans and nature in specific areas.
However, there is a need to improve some parts. Below You could find a few suggestions for improving the paper:
1. The work should be technically checked once again. For example, there is a sign (;) at the end of keywords (Line 23).
2. Hypotheses chapter are missing.
3. Conclusion chapter is also missing.
4. The map (Figure 2) does not provide much information to readers outside the area. We suggest attaching another map instead.
5. Also, I suggest consider following literature:
a. Xu, J., Xu, J., Gu, Z., Chen, G., Li, M., & Wu, Z. (2022). Network text analysis of visitors’ perception of multi-sensory interactive experience in urban forest parks in China. Forests, 13(9), 1451.
b. Zhang, H., Wei, G., Fan, H., Li, J., & Yu, K. (2021, July). Research on the Post Occupancy Evaluation of Urban Park based on internet reviews—Take Saihan Tara City Park in Inner Mongolia as an example. In IOP Conference Series: Earth and Environmental Science (Vol. 825, No. 1, p. 012023). IOP Publishing.
I think that this paper could be a good contribution to theory and practice, on the basis of providing an interesting finding. The relevance of the research realized by the authors of the paper is obvious.
Reviewer 2 Report
Comments and Suggestions for Authors
The topic is relevant to readers. The research methods used provide access to a specific audience - quite large, but limited to those who felt the need to express their opinions online. This is a significant drawback. Qualified tourists may not necessarily lead a life on social media. It is more likely that influencers will share their experiences online. This may exclude the senior adults group. Of course, the method is innovative and based on technological capabilities, but it completely overlooks the feelings of those who do not share their experiences on the Internet. Direct interviews - although more labor-intensive - do not have this drawback. Already at the introduction stage (line 48), indicate the number of visitors and specify whether this is a place of mass tourism - in China, the scale is different than in Europe. In Poland, 2 million tourists annually in Tatra National Park is a lot. I doubt that the number 2 million would impress anyone in China. (line 63) - indicate the number of accidents. Unfortunately, in Europe, a significant number of accidents involving people recording TikTok videos, taking selfies, or unqualified tourists are observed. For example, this year alone, there have been fatal accidents in the Karkonosze National Park known as the "death gully". No qualified tourist would take this route in winter (it is closed). The trend for nature means that unprepared people go to the mountains. Is it the same in China? Does environmental education include rules for behavior in the field? Add a map of a larger area (e.g., all of China) to Figure 2 - readers outside the region do not know where it is (I admit I had to check). I consider it important that the authors combine natural attractions with cultural ones - it is important to perceive the space comprehensively. In the discussion, I miss reference to the problems of mass tourism. Are national parks interested in increasing the number of visitors, or will they rather limit the number of tourists? Can environmental education be conducted in less valuable natural areas? There is a bit of a lack of reference to European literature in the literature.
After the correction, I recommend publication.
